# Nicotinamide Inhibits T Cell Exhaustion and Increases Differentiation of CD8 Effector T Cells

**DOI:** 10.3390/cancers14020323

**Published:** 2022-01-10

**Authors:** Sara Alavi, Abdullah Al Emran, Hsin-Yi Tseng, Jessamy C. Tiffen, Helen Marie McGuire, Peter Hersey

**Affiliations:** 1Melanoma Immunology and Oncology Program, The Centenary Institute, University of Sydney, Camperdown 2050, Australia; s.alavi@centenary.org.au (S.A.); AEMRAN@mgh.harvard.edu (A.A.E.); c.tseng@centenary.org.au (H.-Y.T.); j.tiffen@centenary.org.au (J.C.T.); 2Melanoma Institute Australia, Crows Nest, Sydney 2065, Australia; 3Melanoma Epigenetics Lab, The Centenary Institute, University of Sydney, Camperdown 2050, Australia; 4Cutaneous Biology Research Center, Department of Dermatology, Massachusetts General Hospital, Harvard Medical School, Charlestown, MA 02114, USA; 5Ramaciotti Facility for Human Systems Biology, University of Sydney, Sydney 2050, Australia; 6Faculty of Medicine and Health, School of Medical Sciences, University of Sydney, Sydney 2050, Australia

**Keywords:** T cell exhaustion, nicotinamide, EZH2, TOX, inhibitory receptors, epigenetics, metabolism

## Abstract

**Simple Summary:**

T cells that undergo repeated stimulation undergo a change called T cell exhaustion (TEx) that limits their ability to kill cancer cells and infections with microorganisms. The factors involved in the initiation and kinetics of these changes are poorly understood. In this research the authors have used an in-vitro model of repeated stimulation to investigate whether changes of TEx can be prevented by providing nicotinamide that is known to be integral in energy supply to cells. The model was found to be a good replicate of TEx changes such as upregulation of inhibitory receptors and down regulation of cytokines involved in killing of cancer. The results provide strong support for the notion that TEx results from poor energy supply from mitochondria due to generation of reactive oxygen species and that these changes can be prevented and reversed by treatment with nicotinamide. These findings appear to have important implications for treatment of cancers by immunotherapy.

**Abstract:**

One of the limitations of immunotherapy is the development of a state referred to as T cell exhaustion (TEx) whereby T cells express inhibitory receptors (IRs) and lose production of effectors involved in killing of their targets. In the present studies we have used the repeated stimulation model with anti CD3 and anti CD28 to understand the factors involved in TEx development and treatments that may reduce changes of TEx. The results show that addition of nicotinamide (NAM) involved in energy supply to cells prevented the development of inhibitory receptors (IRs). This was particularly evident for the IRs CD39, TIM3, and to a lesser extent LAG3 and PD1 expression. NAM also prevented the inhibition of IL-2 and TNFα expression in TEx and induced differentiation of CD4+ and CD8 T cells to effector memory and terminal effector T cells. The present results showed that effects of NAM were linked to regulation of reactive oxygen species (ROS) consistent with previous studies implicating ROS in upregulation of TOX transcription factors that induce TEx. These effects of NAM in reducing changes of TEx and in increasing the differentiation of T cells to effector states appears to have important implications for the use of NAM supplements in immunotherapy against cancers and viral infections and require further exploration in vivo.

## 1. Introduction

The introduction of immunotherapy based on immune checkpoint blockade (ICB) on T cells has been a major advance in treatment of cancers such as melanoma and lung cancer and is now considered to be the standard of care. Limitations of these treatments are the relatively high rates of primary and acquired resistance to these treatments. Multiple causes of resistance to immunotherapy have been defined [1,2,3] including defects in antigen presentation [4] and low T cell infiltration into tumors [5]. Of particular interest has been whether resistance to immunotherapy may result from decreased function of T cells associated with a state referred to as T cell exhaustion (TEx) that results from prolonged and repeated stimulation of T cells. TEx cells become dysfunctional with reduced production of cytokines such as IFN-γ, IL-2, and TNF-α and cytotoxic effectors such as granzyme B. Multiple inhibitory checkpoint receptors (IRs) such as PD1, TIM3, LAG3, TIGIT, and 2B4 may be expressed on their surface [6,7]. The changes of TEx appear to be regulated by the transcription factor TOX which is a nuclear DNA binding factor that is upregulated by the nuclear factor of activated T cells (NFAT 2) resulting from chronic T cell receptor activity [8,9]. Recent single cell studies in melanoma have shown that TEx changes are particularly evident on high affinity CD8 T cells against neoantigens and hence potentially a major limitation on the effectiveness of immunotherapy in melanoma [10]. The extent to which TEx is due to epigenetic changes has received much attention in a number of studies [11,12,13,14,15].

Several studies have raised questions concerning the role of metabolism in development of TEx on the premise that the high energy demands of responding lymphocytes may be compromised by competing demands of a growing tumor [16,17] resulting in low glucose levels and low energy output from mitochondria. Nicotinamide adenine dinucleotide (NAD) has long been known as essential for a wide range of biological processes as illustrated by severe deficiency which leads to the development of pellagra characterized by dermatitis, dementia, and diarrhea. Although studied initially for its role in redox reactions of the electron transport chain it is now known to have a much broader role in biology. Several comprehensive reviews detail its synthesis in the body from tryptophan and forms of niacin such as nicotinamide (NAM), nicotinic acid (NA), nicotinamide riboside (NR) and nicotinamide mononucleotide (NMN) as well as its degradation and salvage pathways [18,19].

Recent studies have drawn attention to the importance of NAD in skin cancers and immune responses. In particular a large randomized controlled trial on 386 participants at risk of developing skin cancers found that 500 mg NAM given twice daily for 12 months reduced the rate of non-melanoma skin cancers by 23% compared to placebo [20]. Melanoma arising in five patients on this trial had significant increases in CD4+ and CD8 T cells in peritumoral and tumor infiltrating sites compared to five melanoma in the placebo group [21]. Further evidence for the link to immune function were studies showing that tumour infiltrating lymphocytes (TILs) in melanoma had abnormalities in mitochondria associated with TEx that could be reversed by administration of NR [22].

In the present studies we have adapted an in-vitro model system of TEx [23] to further explore the kinetics of TEx in human lymphocytes and the role that NAD may have in development of the features of TEx. This model allows their evaluation free of conflicting signals from the tumor environment.

## 2. Methods and Materials

### 2.1. Repeated Stimulation of Peripheral Blood Mononuclear Cells (PBMCs) with Anti CD3 and CD28

All samples from human subjects were obtained and experimental procedures were carried out in accordance with the guidelines of local ethical approval requirements, under the National Statement on Ethical conduct in Human Research, NEAF (AU 1/9D50312) and HREC/11/RPAH/32. De-identified blood samples were obtained from healthy donors (*n =* 6) from leukoreduction filters received from the Australian Red Cross Lifeblood research resource. Blood processing, using ficoll-paque purification to separate PBMCs followed standard published protocols [24].

Guided by published methodology for in vitro T cell exhaustion modeling [23], the repeated stimulation model was carried out as shown in the Figure 1A, and Appendix A. Approximately, 1 × 10^6^ cells were suspended in 24 well plates in RPMI (Life Technologies, Carlsbad, CA, USA, cat 11875-093), 5% heat inactivated human serum (Sigma, St. Louis, MO, USA, cat. F4135).

Dynabeads Human T-Activator CD3/CD28 (Life Technologies, 11131D) were used for cell stimulation every other day according to manufactures protocol. In each stimulation, beads were removed, cells were counted and suspended in media with fresh CD3/CD28 dynabeads. For Intracellular detection of cytokines, cells were incubated in cell stimulation cocktail of phorbol 12-myristate 13-acetate (PMA), ionomycin and brefeldin A (Cell Stimulation Cocktail (plus protein transport inhibitors) (500×), Thermo Fisher Scientific, Waltham, MA, USA, cat. 004975-03) for 4 h to activate cells to produce cytokines and block the secretion of cytokines in the endoplasmic reticulum and Golgi apparatus.

For treated cells, the following agents were used: EZH2 inhibitors GSK343 (HY-13500, MedChem express, 5μM), Nicotinamide (NAM) (N0636, Sigma-Aldrich, St. Louis, MO, USA, 20 mM), Selisistat (EX-572 [CAS 49843-98-3], Medchem express, Monmouth Junction, NJ, USA, 2 µM).

### 2.2. Flow Cytometry

PBMCs were incubated for 40 min with surface anti human antibodies in FACS buffer (PBS supplemented with 5% FBS, 10 mM EDTA, and 0.05% sodium azide). All antibodies were titrated prior to experiment to ensure optimal concentrations were used. For gating strategy, fluorescence minus one controls (FMO) were included with each experiment. Cell viability was assessed by staining cells with Live Dead near-IR fixable dye (Invitrogen, Thermo Fisher Scientific, Waltham, MA, USA). Intracellular staining of PBMCs for flow cytometry analysis was performed using intracellular fixation and permeabilization buffer set (Invitrogen, Thermo Fisher Scientific). Cells were incubated in fixation buffer for 40 min. Then cells were incubated for additional 30 min in 1× permeabilization buffer containing intracellular antibodies followed by two-time washes in permeabilization buffer and one-time wash in FACS buffer before flow cytometry analysis.

Comprehensive phenotyping multi-parameter flow cytometry of stained PBMCs was performed with a multicolour BD LSR II flow cytometer (BD Biosciences, San Jose, CA, USA) and the FlowJo software (version 10.6.1 Mac OS X, BD, San Jose, CA, USA) was used for data analysis. Gating strategy is shown in full in Appendix A.

The following antibodies were used for cell staining: PE-Cyanin7 conjugated anti-human CD3 (cat. 557851, Clone.SK7), PerCP-Cy5.5 conjugated anti-human CD4 (cat. 341654, clone. SK3 RUO/GMP), V500 conjugated anti-human CD8 (cat. 561617, clone. SK1), BUV737conjugated anti-human CD279 (PD1) (cat. 612791, clone. EH12.1), Brilliant violet 786 conjugated anti-human IFNg (cat.563731, clone. 4S.B3), Alexa Fluor 700 conjugated anti-human TNFa (cat. 557996, clone. MAb11), BV711 conjugated anti-human IL-2 (cat. 563946, clone.5344.111), PE conjugated EZH2 (cat.562478, clone. 11/EZH2) all of which were purchased from BD Biosciences. Brilliant Violet 605 conjugated anti-human CD39 (cat. 328236, clone. A1), Brilliant Violet 421 conjugated anti-human CD366 (Tim-3) (cat. 345008, clone. F38-2E2), PE/Dazzle 594 conjugated anti-human CD223 (LAG-3) (cat. 369331, clone. 11C3C65) were purchased from Bioledgend. eFluor660 conjugated TOX (cat. 50-6502-82, clone. TXRX10) was obtained from Thermo Fisher Scientific (Waltham, MA, USA). Alexa Fluor (red) 647 conjugated Tri-Methyl-Histone H3 (Lys27) (cat. 12158S, clone. C36B11, Cell Signalling Technology, Danvers, MA, USA).

### 2.3. Mitochondrial Potential and Mitosox ROS Studies

These studies were carried out as described in [25]. Mitochondrial superoxide was analyzed using MitoSOX Red Mitochondrial Superoxide Indicator (cat. M36008, Thermofisher Scientific) at 5µM according to the manufacturer’s protocol. Mitochondrial mass was analyzed using Mitotracker Green FM (cat. M7514, Thermo Fisher Scientific, Waltham, MA, USA) at 100 nmol/L according to the manufacturer’s protocol. Mitochondrial membrane potential was analyzed using tetramethylrhodamine methyl ester (TMRM; cat. M22426, Thermofisher Scientific) at 10 nmol/L according to the manufacturer’s protocol.

Briefly, cells were incubated at 37 degrees in 10 µM Oligomycin for 2.5 h in the before staining for 20 min in pre-warm PBS + 2%FBS containing Mitotrackers or Mitosox. Live dead Fixable Aqua Dead Cell Stain Kit (L34957, Thermo Fisher Scientific) was used to determine cell viability. Then cells were washed two times in buffer before acquisition on ARIA flow cytometer (BD Bioscience), with data analysis performed using FlowJo software (version 10.6.1 Mac OS X, BD, USA).

### 2.4. Mass Cytometry

PBMCs were stained for mass cytometric analysis according to previously published methods [24]. All antibodies were validated, pre-tittered and supplied in per-test amounts by the Ramaciotti Facility for Human Systems Biology Mass Cytometry Reagent Bank. A detailed list of antibodies and corresponding metal tags is provided in Appendix A. Reagent bank antibodies were purchased in a carrier-protein-free format and conjugated by the Ramaciotti Facility for Human Systems Biology with individual metal isotopes using the MaxPAR conjugation kit (Fluidigm, South San Francisco, CA, USA) according to the manufacturer’s protocol. Data in FCS3 file format were collected using the Helios software (V6.3.119) on a CyTOF 2 Helios upgraded mass cytometer (Fluidigm, South San Francisco, CA, USA) and normalized across experiments using EQ Four Element Calibration Beads using data processing function within the acquisition software. FlowJo software was subsequently used to pre-gate on DNA+, live, CD45+ cells, then to gate all cells into major immune cell populations.

The t-stochastic neighborhood embedding (t-SNE) algorithm was applied using the FlowJo Auto (opt-SNE) function, on FlowJo 10.6.1 Mac OS X. All fcs files from across day seven experimental conditions were gated to the level of live cells with stringent doublet exclusion and concatenated to a single file. The t-SNE algorithm was run using the full panel of markers (excluding CD45 barcoding, viability and non-T cell markers) with algorithm conditions listed here; Iterations: 1000, Perplexity: 30, Learning rate (eta) 24218, KNN algorithm exact (vantage point tree), Gradient algorithm (barnes-hut).

### 2.5. Statistical Analysis

Statistical analysis and graphical representation were performed using Graphpad Prism, version 7.0 (GraphPad Software, Inc., San Diego, CA, USA). Group comparisons were performed as detailed in the figure legends.

## 3. Results

### 3.1. Nicotinamide (NAM) Inhibits the Expression of Inhibitory Receptors (IRs) during Repeated Stimulation

This study set out to assess the role metabolism has in development of TEx, as mediated through NAM. Utilizing an established in vitro model of chronic stimulation [23], as shown in Figure 1A, we demonstrated an expected increase in PD1 expression with repeated stimulation with anti CD3 and anti CD28 along with the other IRs TIM3, LAG3 and CD39, indicative of TEx (Figure 1B). This applied to both CD4+ and CD8+ T cell subsets. Addition of NAM in the cultures reduced PD1 upregulation in the CD4+ T cells compared to untreated conditions (statistical significance reached using the Holm-Sidak method, with α = 0.05 for days two to nine-fold change relative to untreated). The expression of TIM3 was almost completely inhibited at all time periods in both subsets. NAM did not consistently modify LAG3 expression compared to expression in control condition in CD4+ or CD8+ T cells (Figure 1C). Expression of CD39 was strongly inhibited in CD4+ T cells by day nine and earlier in CD8+ T cells by NAM. Downregulation of the IRs by NAM was consistent across four biological repeats, from independent experiments as shown by Figure 1C where the CD8+ T cell data from days five, seven, and nine has been normalized as fold change relative to untreated cultures, from the equivalent day in order to account for marker expression differences between donors.

We examined whether NAM had selective effects on expression of particular IRs. The main finding was that by day nine IR expression in untreated conditions were dominated by CD39 expression (62% of IR+ cells, data not shown) but this was much reduced in the presence of NAM (31% of IR+ cells for NAM only) consistent with the strong downregulation by NAM.

Given that Sirtuins are activated by NAD [26] we examined whether activation of Sirtuins may be involved in the inhibition of TEx by a repeat of the study with the SIRT1 inhibitor Selisistat. This had no effects on the results which indicated that NAM was not acting by activation of SIRT1 (Appendix A).

### 3.2. NAM Reverses the Inhibitory Effects of TEx on Intracellular Production of IL-2, IFNγ and TNFα

In previous studies in this model EZH2 was reported to have selective effects on inhibiting IL-2 production [23] suggesting that IR and cytokine expression may be regulated differently in TEx. To further examine this, we studied intracellular cytokine production in presence of NAM. As shown in Figure 2A NAM produced marked increases in production of TNFα in CD8+ T cell subsets compared to time dependent reduction in control condition. NAM induced changes in IL-2 were highly variable across donors (Figure 2B). Increases in IFNγ were not as marked in presence of NAM, generally tracking below control levels, however this only reached significance on day two in CD4+ T cells (using the Holm-Sidak method, with α = 0.05 on fold change relative to untreated). Previous studies have implicated EZH2 in regulation of TOX [25] and in view of this we included the EZH2 inhibitor GSK343 in the stimulation model as described in Figure 1A. The EZH2 inhibitor had minimal effect on cytokine production except for small increases in IL-2 production (data not shown).

We also examined production of Granzyme B which is part of the CD8+ T cell effector response. Using high-dimensional mass cytometry we showed that TNFα expression was strongly associated with co-expression of granzyme B in untreated cultures but was downregulated in NAM cultures (Figure 2C). Figure 2D shows that overall NAM increased the capacity for TNFα cytokine expression in all of the studies and IL-2 in two studies from day four to five onwards. The effects on IFNγ were relatively minor.

Given the heterogeneity in IR expression we examined whether expression of particular IRs was associated with more or less cytokine production. As shown in Figure 2D NAM appeared to maintain more multi cytokine production on CD39 expressing cells compared to that in TIM3 expressing cells on day six to seven, even though these IR were downregulated to similar extent by NAM. Similar results were found in three repeats of these studies. In studies by others TIM3 was associated with TCF1 expressing T cells [27] but this was not examined in the present studies.

### 3.3. NAM Inhibits the Production of TOX Irrespective of Its Effect on EZH2 Mediated H3K27 Trimethylation

In view of the evidence linking TOX expression to the TEx program we examined the effect of adding NAM on TOX expression. As shown in Figure 3A TOX protein levels on day six to seven and eight to nine were markedly downregulated compared to control untreated cultures. In view of reports that EZH2 down regulated TOX [25] we examined the effect of inhibiting EZH2 with the GSK343 inhibitor. It is important to note that there was a trend for the maintenance of H3K27 methylation levels in NAM conditions, however TOX levels in both subsets remained inhibited.

### 3.4. NAM Induces Differentiation of CD4+ and CD8+ T Cells into Effector Differentiation States

We reasoned that the differences in expression of inhibitory receptors (IRs) over time during treatment with NAM may have been due to expression of different differentiation states. To answer this, we studied differentiation into T precursor naïve (CD45RA+, CD27+, CCR7+), T central memory (T CM) (CCR7+, CD27+) T effector memory (T EM) (CCR7-_ve), and T effector terminally differentiated (T EMRA) (CCR7-ve CD45+) cells at each stimulatory time period as shown in Figure 4A. Studies with the EZH2 inhibitor were included as a comparator as EZH2 is known to maintain poly functional states [27]. The results showed that NAM increased T EM and T EMRA from day four to five in both CD4+ and CD8+ T cells. In contrast EZH2i increased the T CM cells at these times particularly in CD8+ T cells. Adding the EZH2i to NAM did not reduce the effects of NAM on differentiation except in TIM3 expressing cells where there was a decrease of NAM’s effect on differentiation.

We asked whether IR expression may also influence the differentiation state. As shown in Figure 4B in the day six to seven cultures CD8+ T cells expressing LAG3 or TIM3 had higher levels of effector memory (T EM) and T effector terminally differentiated (TEMRA) than PD1 expressing CD8+ T cells. CD39 expressing CD8+ T cells had less T EMRA cells. Again, this contrasted with the effect of GSK343 which increased T CM in all subsets particularly in LAG3 expressing CD4+ and CD8+ T cells consistent with the view that EZH2 was involved in differentiation of T cells [28,29]. Induction of T EMRA was less in CD4+ T cells except in LAG3 expressing CD4+ T cells. Notably, there was no selective enrichment or depletion of Treg CD4+ T cells under the assessed treatment conditions (Figure 4C). The decrease in CD39 expression observed across all CD4+ T cells in NAM cultures (Figure 1B) was also evident specifically for assessment of Tregs.

To further illustrate the influence that NAM had on the expression of IRs and differentiation of the T cells a tSNE analysis was carried out on day seven data, Figure 4D,E. As shown in Figure 4E, NAM markedly suppressed the CD39 and Tim3 IRs compared with control untreated with less effects on PD1 and LAG3. In contrast the EZH2 inhibitor (GSK343) maintained high levels of IR expression consistent with a role in suppression of the IRs. The figure also confirms that the effects of NAM were independent of EZH2i. As shown in column five in Figure 4E NAM was associated with differentiation into the TEMRA state whereas GSK343 maintained or decreased their differentiation.

### 3.5. NAM Reverses TIM3 and LAG3 Expression in Established TEx Cells

Although NAM was able to prevent the development of TEx features during repeated stimulation it was not clear whether it would reverse pre-existing changes of TEx. To answer this T cells were stimulated to create the onset of TEx and NAM was added at the time of the third repeat stimulation (day six to seven) when the TEx markers were well established (Figure 5A). Significant reversal of TEx induced IR PD1 and TIM3 for CD4+ T cells and LAG3 across both subsets were seen but reduction of CD39 was minimal (combined data in Figure 5B, with Holm-Sidak α = 0.05 determined significance when assessed as fold change relative to untreated for each donor tested). Similarly, treatment with NAM on day six to seven increased production of IFNγ, (reaching significance in CD8+ T cells by Holm-Sidak determination of fold change relative to untreated) but was unable to recover TNFα levels as observed when treated earlier (Figure 5C). TOX expression was further inhibited by NAM in both CD4+ and CD8+ subsets (Figure 5D). Taken together this is evidence that addition of NAM to T cells that have already developed TEx can reverse all but the expression of CD39. Whether a longer follow up time would reduce CD39 is unknown. Although not shown, we found again as shown in Figure 3A that EZH2 function in terms of H3K27me3 levels was increased by NAM but the EZH2i did not reverse the effects of NAM on TOX levels indicating that EZH2 was not involved in TOX regulation.

### 3.6. NAM Reduces Reactive Species Generated in the Repeated Stimulation Model

To examine ROS production, we treated unfixed repeatedly stimulated CD3 T cells with mitoSOX as described elsewhere [30]. As shown by Figure 6A there was an initial increase in ROS by day two which was maximal on days five to seven and then largely sustained at day nine. As a comparison, cultures with n-acetyl cysteine (NAc) were added as an antioxidant. A trend towards reduced ROS levels shown in NAM treated cultures was associated with similar values to NAC initially at day two, with a marked reduction compared to untreated day nine culture conditions.

Measurement of mitochondrial membrane potential (MMP) by mitotracker deep red (MDR) showed comparable levels to the untreated cultures with small increases at day seven and nine. (Figure 6B). Assessment of the mitochondrial mass by mitotracker green (MTG) staining showed that treatment with NAM was associated with trend to decrease the mitochondrial mass compared to that in untreated cultures on days two and five, and significant increase on day nine (Figure 6C). This may reflect NAM induced reduction in mitochondrial damage plus mitophagy and clearance of damaged mitochondria as described by others [31].

When the results were presented as a ratio of MDR/MTG there was an increase in mitochondrial potential in NAM treated cultures on day two above the untreated cultures but similar ratios on days five, seven, and nine (Figure 6D). The increase at day two coincides with the maximal decrease in ROS levels on day two induced by NAM. The results are therefore consistent with a decrease in ROS due to NAM and a consequent reduction in mitochondrial damage.

This interpretation of the data is shown in Figure 6E showing that NAM reverses ROS induced TOX resulting in decreased expression of IRs and increased production of cytokines TNFα. NAM however has the potential to increase EZH2 and thereby complement the effects of NAM in inducing T cell differentiation and maintenance of their polyfunctionality.

## 4. Discussion

This study confirms that the model of repeated stimulation of human lymphocytes with anti CD3 and anti CD28 described by others [23], reproduces the changes in inhibitory receptors and cytokine production described in T cell exhaustion (TEx) states. Our focus has been to examine whether the detailed kinetic and immunologic study possible in this model can provide further insights into the mechanisms underlying development of TEx that can be exploited in treatment of cancers. Our findings that nicotinamide (NAM) can inhibit and reverse many of the features of TEx highlight the possible importance of metabolic changes as the driver of T Ex states. The premise being that high energy demands of immune responses against cancer are competed for by the cancer cells and this results in mitochondrial dysfunction that triggers the epigenetic changes associated with the TEx state [16,17]. NAM could be involved in these processes by its key role as a precursor of NAD^+^/NADH that is involved in many oxidation -reduction reactions such as ATP production in the Krebs cycle [19]. NAD can also activate Sirtuins [26] but the SIRT1 inhibitor Selisistat did not inhibit NAM activity making it unlikely to be involved in downregulating TEx.

Given this background it was of great interest to see that addition of NAM in the repeated stimulation cultures prevented the induction of IRs on both CD4+ and CD8+ T cells and strongly increased TNFα and IL-2 cytokine production with less effect on IFNγ production. Reversal of LAG3 expression was partial rather than complete. The inhibition of PD1 was also incomplete indicating different signals were involved as reviewed elsewhere [32]. PD1 is expressed on all T cells during activation and is not necessarily a marker of TEx [33,34]. T cell receptor (TCR) engagement and activation of nuclear factor of activated T cells (NFAT) is the principal activator but other transcription factors such as FOXO1, T-bet, and BLIMP1 are believed to be involved [33,35].

In this model TOX protein levels increased from day four with maximum levels on day eight. Addition of NAM in the cultures was associated with approximately a 50% reduction in TOX levels in CD4+ and CD8+ T cells after day four. This questioned whether NAM may decrease the TEx markers by downregulating TOX and if so by what mechanism. In this context much attention has been given to the role of EZH2 which is the catalyst in the PRC2 complex. Previous studies have shown that EZH2 has important roles in determining the differentiation state of CD8+ T cells and maintenance of a polyfunctional state. Studies on the kinetics of H3K27me3 mediated by EZH2 revealed that the trimethylation state persisted at high levels when NAM was present whereas there was a slow decline in the cultures without NAM. These results raised questions as to whether NAM down regulation of TOX may have been mediated by EZH2 but when inhibitors of EZH2 were included in cultures with NAM TOX remained downregulated. This implied that NAM was regulating EZH2 but also had other effects that were limiting TEx.

For this reason, attention was given to NAMs role in generation of reactive oxygen species (ROS) that has been implicated in several studies as a cause of TEx [36,37]. The present studies were consistent with this in that NAM reduced ROS levels almost to the same extent as the antioxidant n-acetylcysteine (N-AC). These results were also consistent with previous studies showing that NR can increase the polarization state of mitochondria [22]. Taken together these results are consistent with the idea that NAM increases energy levels by oxidative phosphorylation (OXPHOS) and triggers downregulation of ROS mediated regulation of TOX.

The role of EZH2 in TEx however requires further study. Inhibition of EZH2 downregulated the effects of NAM on LAG3 expression but did not appear involved in NAM downregulation of TIM3 or CD39. We were particularly interested in LAG 3 expression as clinical studies are pointing to blockade of this receptor as adding significant benefit to blockade of PD1 [38,39]. It is of interest that in single cell studies TOX knockdown did not totally reverse LAG3 expression [40] so that EZH2 inhibition may have some role in regulation of LAG3 that requires further study. Differentiation into T EM and T EMRA effector cells was more marked in the LAG3 and TIM3 CD8+ T cells. This increased differentiation may have involved EZH2 in that inhibition of EZH2 maintained the T cells in T CM states. The NAM induced differentiation into more effector phenotypes may also account for the disparity of increased TNFα but reduced granzyme B expression in that granzyme B may have been lost by degranulation from the more mature CD8+ T cells [27].

Although NAM appeared to inhibit the development of TEx in this model it was not clear whether it would reverse pre-existing changes of TEx. To answer this NAM was added at the time of the third repeat stimulation (at day six to seven) when the TEx markers were well established. Addition of NAM to pre-existing TEx cells downregulated TOX and partially restored cytokine production. Significant reversal of TEx markers TIM3 and LAG3 were seen but reduction of CD39 was minimal. This may indicate that upregulation of CD39 is maintained by mechanisms set in train by TOX but not maintained by TOX. CD39 is an ectoenzyme that degrades ATP [41] so it is possible that any increase in ATP from NAM treatment may be degraded by CD39 expressing cells. CD39 was retained longer than PD1 in other studies using CD3/CD28 co stimulation [42]. The retention of CD39 on TEx cells may be relevant in explaining the association of CD39 expressing CD8+ T cells with poor outcomes in patients treated with anti PD1. Taken together this is evidence that NAM may not only prevent TEx but also reverse some of the features of established TEx.

In summary the repeated stimulation model has provided strong support for the thesis that T cell exhaustion is triggered by metabolic factors that can be reversed by addition of NAM to the cultures. NAM was able to downregulate ROS levels in mitochondria which is consistent with previous studies suggesting that ROS generation by mitochondria was responsible for upregulation of TOX and TEx. Importantly NAM increased the differentiation of CD8+ T cells into T EM and T EMRA effector cells that is relevant to CD8+ T cell function against cancers and viral infections. Addition of NAM to T cells with established TEx showed downregulation of TOX and reversal of most TEx features with the exception that suppression of cytokines in CD39 expressing cells was not reversed by NAM. It is possible this may be from degradation of ATP required by the CD8+ T cells by the CD39 ectoenzyme. NAM has received attention for its biological effects in a wide range of processes such as aging, cardiac and neurological disorders. The present study strengthens its potentially important role in immune responses that needs further exploration in vivo.

## 5. Conclusions

In conclusion, we have described herein studies on human T cells showing that nicotinamide can prevent and reduce the features of T cell exhaustion that occur in T cells undergoing repeated stimulation. T cell exhaustion can be a major limitation to the effectiveness of immunotherapy so that the results have significance over a broad range of immunotherapy treatments. The studies show the kinetics of development and utilize the latest in-vitro analytical techniques to describe the effects of nicotinamide on features of T cell exhaustion.

## Figures and Tables

**Figure 1 cancers-14-00323-f001:**
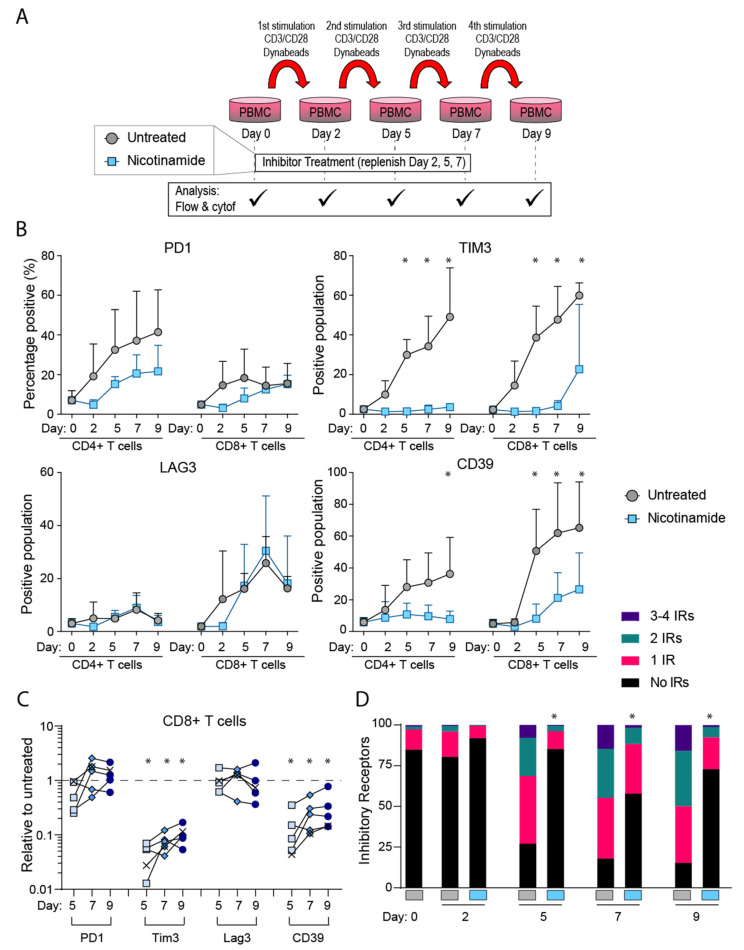
Expression of IRs in vitro in the presence of NAM. NAM reduced expression of TIM3 and CD39. (**A**) Schematic showing experimental set up. (**B**) Time course shown for average across four flow cytometry experiments, with error bars indicating standard deviation, and statistical significance between treated and untreated indicated by * as determined using the Holm-Sidak method (α = 0.05). Additional statistics performed on (**C**) combined data when compared to untreated only for each time point, on days five to nine across *n* = 5 in vitro stimulation experiments, Mann-Whitney test (*p* < 0.05 indicated by *). (**D**) Cumulative IR expression of total CD8+ T cells as scored for representative experiment shown in 1A, chi-squared analysis (* *p* < 0.05). Data points marked in (**C**) as ‘X’ were obtained from mass cytometry analysis, with remainder from flow cytometry.

**Figure 2 cancers-14-00323-f002:**
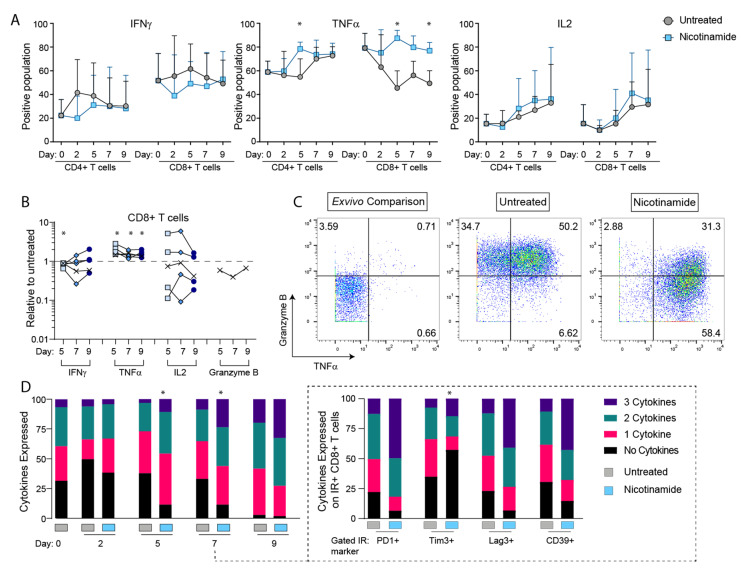
Intracellular cytokine production in the continues presence of NAM. NAM induces higher production of TNF and IL2 in both CD4+ and CD8 T cells and higher production of IFN in CD8+ T cells. (**A**) Time course shown for average across four flow cytometry experiments, with error bars indicating standard deviation, and statistical significance between treated and untreated indicated by * as determined using the Holm-Sidak method (α = 0.05). Additional statistics performed on (**B**) combined data when compared to untreated only for each time point, on days five to nine across *n =* 5 in vitro stimulation experiments, mann-whitney test (*p* < 0.05 indicated by *), and (**C**) representative plots showing difference in granzyme B and TNFα expression patterns. (**D**) Cumulative cytokine expression of total CD8+ T cells, and (**D**) inset IR expressing CD8+ T cells as scored for representative experiment shown in 2A, showing significance by chi-squared analysis (* *p* < 0.05). Data points marked in (**B**) as ‘X’ were obtained from mass cytometry analysis, with remainder from flow cytometry.

**Figure 3 cancers-14-00323-f003:**
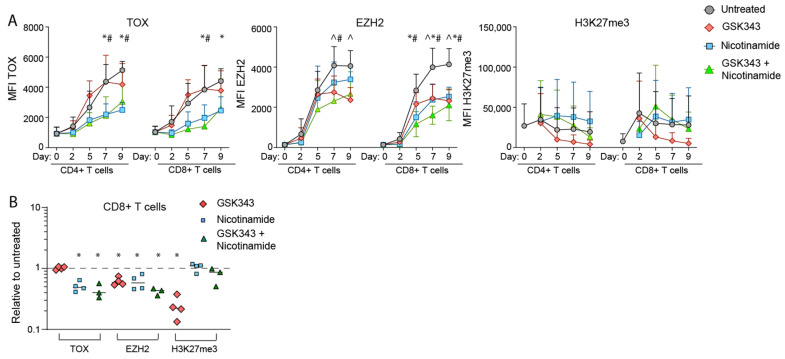
Expression pattern of TOX, EZH2 and H3K27me3 in the presence of NAM and EZH2i. (**A**) Time course shown for average across four experiments, with error bars indicating standard deviation, and statistical significance between treated conditions and untreated indicated by ^ for GSK343, * for nicotinamide and # for GSK343 + nicotinamide, as determined using the Holm-Sidak method (α = 0.05). Additional statistics performed on (**B**) combined data when compared to untreated only for day seven across *n =* 4 in vitro stimulation experiments (*p* < 0.05 indicated by *). Mean fluorescent intensity (MFI) were calculated for each time point.

**Figure 4 cancers-14-00323-f004:**
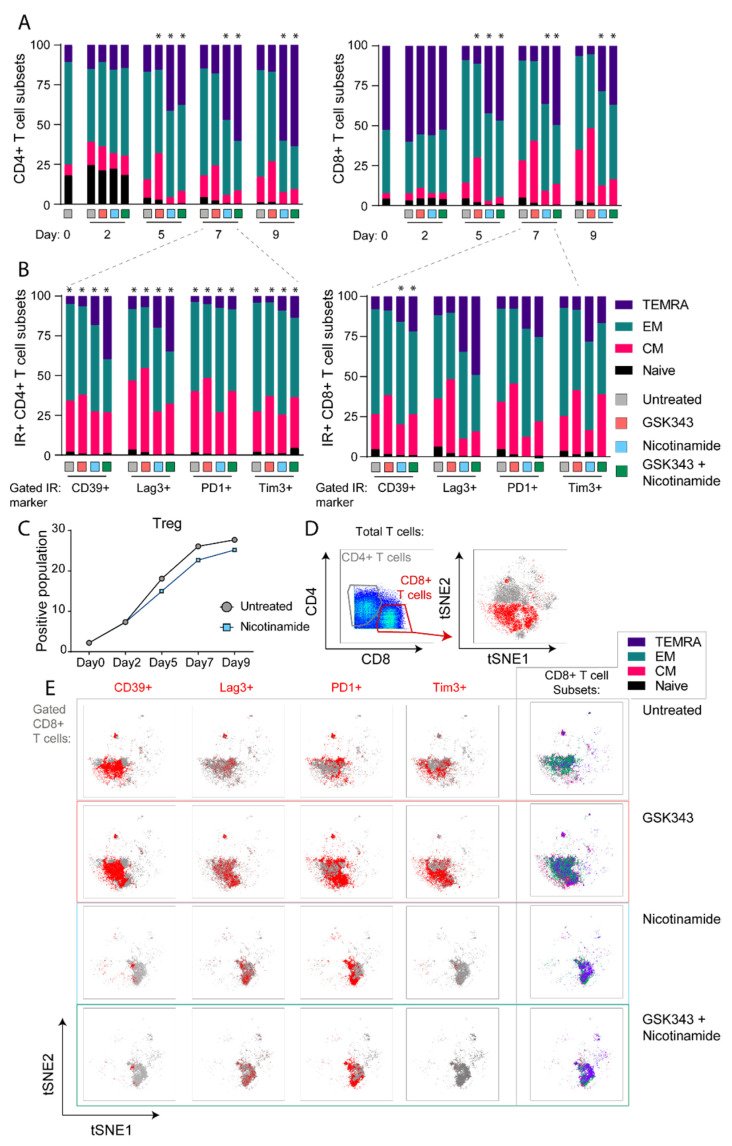
The effects of EZH2i and NAM in T cell differentiation states. (**A**) GSK343 treatment largely matches untreated subsets as tracked through in vitro stimulation time course, with significant increase in central memory subset on day five. NAM drives a high TEMRA proportion in both CD4+ and CD8+ T cells. (**B**) T cells were gated to IR+ cells and assessed for proportions of differentiation subset proportions, on day seven. Assessed by chi-squared analysis (* *p* < 0.05). TEMRA, T effector terminally differentiated; EM, effector memory; CM, central memory. (**C**) Treg phenotype tracked through treatment groups by mass cytometry (CD25+ CCR4+ of CD4+ T cells). (**D**) Representative plot showing T cells on day seven gated for CD4+ and CD8+ and as a tSNE representation, with CD8+ T cells highlighted in red, and of the CD8+ T cells, (**E**) positioning of IR+ cells (in red) and subsets (indicated by color key: TEMRA purple, EM aqua, CM pink and naïve black) across ex vivo comparison and the range of in vitro treatment conditions.

**Figure 5 cancers-14-00323-f005:**
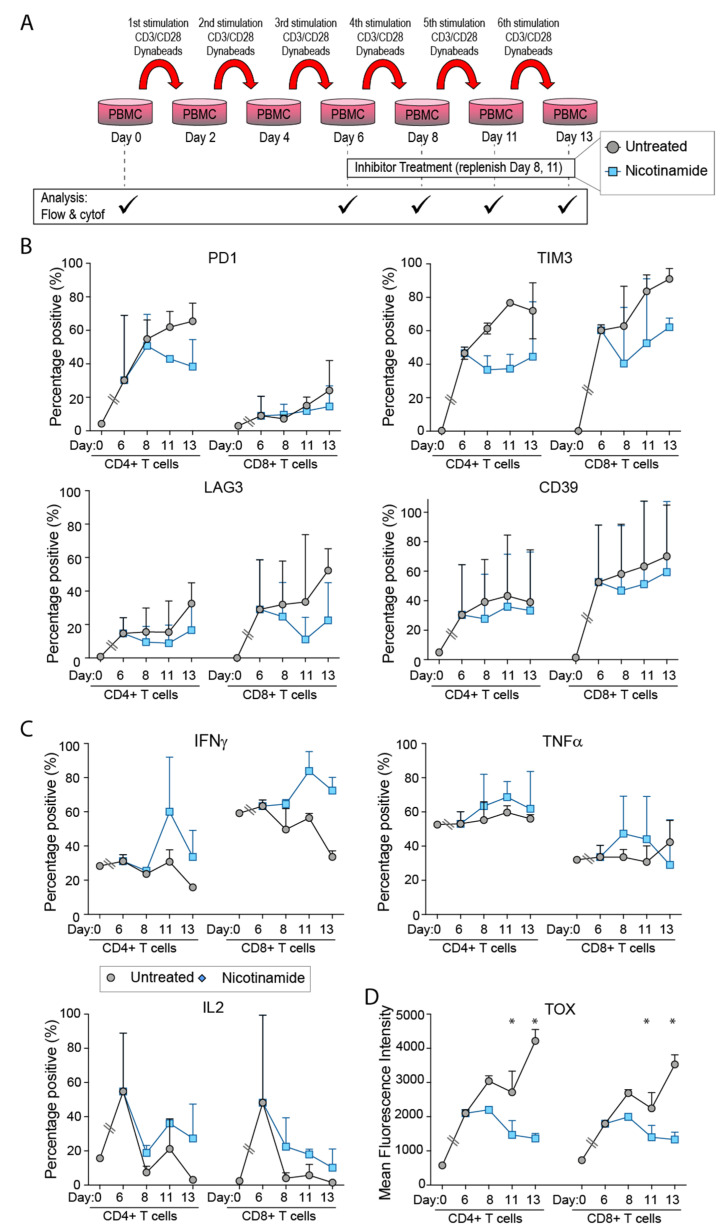
The effects of EZH2i and NAM in established exhausted T cells. NAM treatment down regulates TOX, induces decrease in the expression of TIM3 and LAG3 and increases in the production of IFN and IL-2. (**A**) Schematic showing experimental set up. Time course shown for average across independent experiments, whereby CD3/CD28 bead stimulation until day six was without inhibitor treatment (black line) with inhibitor treatment concurrent with CD3/CD28 bead stimulation thereafter, showing (**B**) Means and standard deviations of expression of IRs, (**C**) cytokines and (**D**) transcription factors as per Figure 1, Figure 2 and Figure 3. TOX, EZH2 and H3K27me3 mean fluorescent intensity (MFI) were calculated for each time point. Statistical significance between treated and untreated indicated by * as determined using the Holm-Sidak method (α = 0.05).

**Figure 6 cancers-14-00323-f006:**
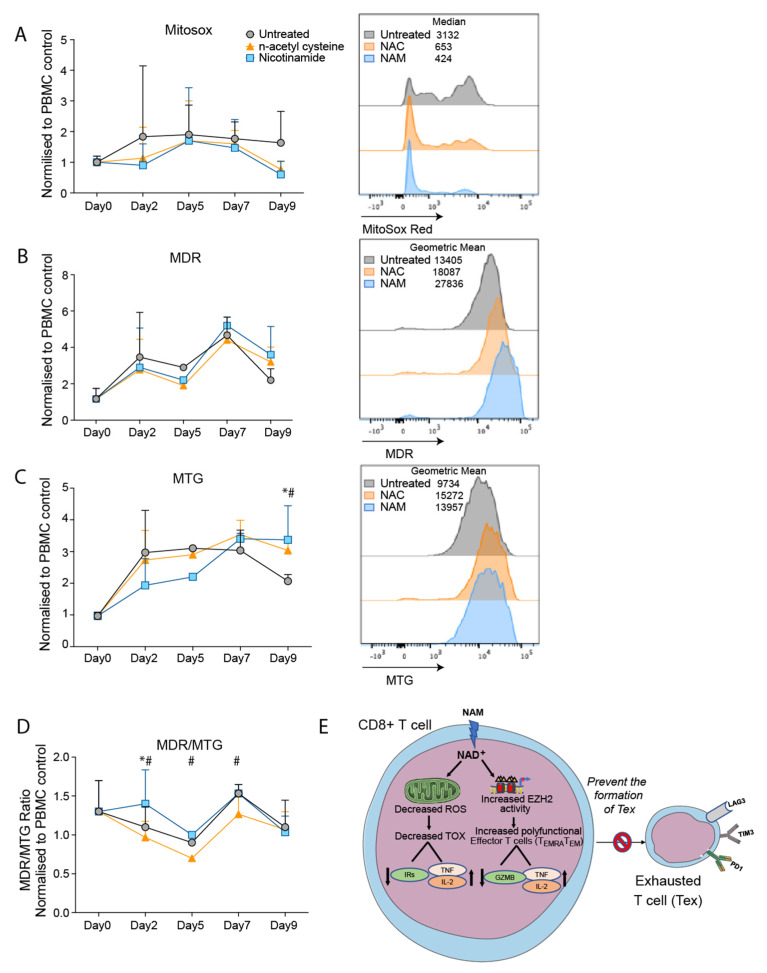
Evaluation of mitochondrial function in the presence of NAM and EZH2i and their combination. (**A**) median expression of MitoSOX based flow cytometry detection of mitochondrial reactive oxygen species (ROS) production, showing mean and standard deviation from three independent experiments. (**B**) Mitotracker red (MDR) and (**C**) green (MTG) assessment of mitochondrial membrane potential and mitochondrial mass, (**D**) with ratio of MDR to MTG also shown for three independent experiments. Statistical significance determined when treated conditions compared to untreated for each donor as indicated by * for nicotinamide and # for n-acetyl cysteine, as determined using the Holm-Sidak method (α = 0.05). (**A**–**C**) graphs are shown as relative to PBMC controls run concurrently for each time point, with histogram showing representative expression comparing treatment groups. (**E**) Schematic view of the effect of NAM on exhausted t cells through mitochondrial function. NAM, nicotinamide; NAC, n-acetyl cysteine.

## Data Availability

The data underlying this article will be shared on reasonable request to the corresponding author.

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
