# Peer review of "Nicotinamide Inhibits T Cell Exhaustion and Increases Differentiation of CD8 Effector T Cells"

_cancers, 2022, doi:10.3390/cancers14020323_

Round 1
Reviewer 1 Report
Authors have removed the EZH2 data and focused on the effects of NAM, which has been a good factor to improve the quality of the paper.
However, there are still some overstatements or descriptions in the text that are then not shown in the figures and that should be corrected.
Examples:
- When describing Fig.1 the authors say that PD-1 up regulation is prevented and that LAG3 expression in CD4+ is increased with NAM (line 200).
- Line 222: ‘As shown in figure 2A NAM produced marked increases in production of IL-2…'
- Line 256: ‘Its is important to note that NAM upregulated H3K27 methylation and that this was inhibited by the GSK343 inhibitor’
- Line 317: ‘ Treatments with NAM on day 6-7 increased production of IFNg and TNFa’ - Line 354: ‘ Assessment… showed that treatment with NAM was associated with marked decreased in mitochondrial mass…' - The final conclusion should also be toned down. Particularly when the authors state that NAM also increases EZH2 (not shown).
Author Response
Authors have removed the EZH2 data and focused on the effects of NAM, which has been a good factor to improve the quality of the paper.
However, there are still some overstatements or descriptions in the text that are then not shown in the figures and that should be corrected.
Examples:
- When describing Fig.1 the authors say that PD-1 up regulation is prevented and that LAG3expression in CD4+ is increased with NAM (line 200).
We thank the reviewers for their comments, double checking statistics thoroughly and have revised any ambiguous text. With regard to this example, we have clarified how significance was reached for PD1 and on review realised LAG3 comment was unnecessary. Text has been modified accordingly.
- Line 222: ‘As shown in figure 2A NAM produced marked increases in production of IL-2…'
We acknowledge that the NAM induced changes in IL-2 where highly variable across donors, and have adjusted the text accordingly.
- Line 256: ‘Its is important to note that NAM upregulated H3K27 methylation and that this wasinhibited by the GSK343 inhibitor’
We revised this text to reflect the findings of the figure more accurately.
- Line 317: ‘ Treatments with NAM on day 6-7 increased production of IFNg and TNFa’
We have clarified in the text how significance was reached for IFNg, and on review realised that the TNFa comment was unnecessary, adding how this compared to earlier finding.
- Line 354: ‘ Assessment… showed that treatment with NAM was associated with markeddecreased in mitochondrial mass…'
After clarifying these findings, we found this early decrease was a trend and have adjusted the text accordingly.
- The final conclusion should also be toned down. Particularly when the authors state that NAMalso increases EZH2 (not shown).
We have taken the reviewers comment on board and have adjusted the text accordingly.
Reviewer 2 Report
In this revised version, the authors have much improved the manuscript, providing more details, and removing some incomplete evidence that risked to lead to overstatements. They have also validated the impact of NAM on known markers of T cell exhaustion in the current model of chronic PBL activation.
Unfortunately, there are still a few points that require attention.
Data from figure 2A and figure 5 are still not consistent regarding IL-2 expression (increase in Figure 2A over time while decrease in Figure 5 over time), whereas the protocol is the same during the first 8 days of culture. The authors state in their response, “the mean of all 3 studies is at baseline” as shown in figure 2B for NAM effect on IL-2 production. Thus the authors cannot state that NAM is inducing an upregulation in IL-2. This should be toned down or explained.
Line 222: “As shown in figure 2A NAM produced marked increases in production of IL-2 and TNFα from both T cell subsets.” Statistical differences in TNFa for CD8 seems validated in Figure 2B, however by visual comparision of the curves NAM doesn’t seem to impact TNFa in CD4+ T cells or IL-2 production in either T cell subtypes. Statistical comparison would be required to validate the point made by the authors. In addition, data presented by the authors from Figure 2B suggest that NAM doesn’t affect consistently IL-2 production, so the statement on IL-2 upregulation upon NAM is incorrect.
Statistics are missing for all data of figure 5 and 6. Mean and SD annotation is not sufficient. The authors should validate that the differences between the treatment groups at the different timepoints are statistically significant in order to draw conclusions.
In figure 5, TOX representation as MFI convincingly demonstrates the effect of NAM, but the question still remains of why only 5-10% of the cells are TOX positive in the current model. This point should be addressed in the discussion.
In figure 6, the authors should perform statistical analysis at least at the different timepoints to validate that there are statistical differences between the treatment groups. Without these proper analyses, the authors cannot draw a firm conclusion that that effects of NAM are linked to regulation of ROS.
Together, conclusions drawn on the impact of NAM on cytokine secretion, ROS or mitochondrial state appear weak in the absence of proper statistical analyses.
Author Response
In this revised version, the authors have much improved the manuscript, providing more details, and removing some incomplete evidence that risked to lead to overstatements. They have also validated the impact of NAM on known markers of T cell exhaustion in the current model of chronic PBL activation.
Unfortunately, there are still a few points that require attention.
Data from figure 2A and figure 5 are still not consistent regarding IL-2 expression (increase in Figure 2A over time while decrease in Figure 5 over time), whereas the protocol is the same during the first 8 days of culture. The authors state in their response, “the mean of all 3 studies is at baseline” as shown in figure 2B for NAM effect on IL-2 production. Thus the authors cannot state that NAM is inducing an upregulation in IL-2. This should be toned down or explained.
We thank the reviewer for the careful response. We found IL2 expression was highly variable across the donors used in our study, and as in the response to reviewer 1 we have stepped back from drawing too many conclusions from the NAM effect on IL2. This may be explainable by a factor we are unaware of (such as CMV sero-status) but given our donors were sourced from the Red Blood bank we do not have this information to correlate to. Furthermore, time course dynamics were highly donor related, with half of our donors observing a decrease in IL2+ CD8+ T cells between day 7-9 (figure 2), however unfortunately, unknown to us choosing, all donors showed this dynamic of decreased IL2 for day 6-8 transition (figure 5).
Line 222: “As shown in figure 2A NAM produced marked increases in production of IL-2 and TNFα from both T cell subsets.” Statistical differences in TNFa for CD8 seems validated in Figure 2B, however by visual comparision of the curves NAM doesn’t seem to impact TNFa in CD4+ T cells or IL-2 production in either T cell subtypes. Statistical comparison would be required to validate the point made by the authors. In addition, data presented by the authors from Figure 2B suggest that NAM doesn’t affect consistently IL-2 production, so the statement on IL-2 upregulation upon NAM is incorrect.
Additional statistics have been added to Figure 2A, and as also requested by reviewer 1, text has been updated to reflect this figure more accurately.
Statistics are missing for all data of figure 5 and 6. Mean and SD annotation is not sufficient. The authors should validate that the differences between the treatment groups at the different timepoints are statistically significant in order to draw conclusions.
Additional statistics have been added to Figures 5 and 6, text has been updated to reflect the figures more accurately.
In figure 5, TOX representation as MFI convincingly demonstrates the effect of NAM, but the question still remains of why only 5-10% of the cells are TOX positive in the current model. This point should be addressed in the discussion.
They may have been some misunderstanding from original manuscript version Figure 5. In fact, nearly all cells express some level of TOX, as shown in supplementary figure 2, thus the reason the MFI metric is most suitable.
In figure 6, the authors should perform statistical analysis at least at the different timepoints to validate that there are statistical differences between the treatment groups. Without these proper analyses, the authors cannot draw a firm conclusion that that effects of NAM are linked to regulation of ROS.
Additional statistics have been added to Figure 6, and text has been updated to reflect this figure more accurately.
Together, conclusions drawn on the impact of NAM on cytokine secretion, ROS or mitochondrial state appear weak in the absence of proper statistical analyses.
We thank the reviewer, and think the changes we have made have adequately addressed their concerns.
This manuscript is a resubmission of an earlier submission. The following is a list of the peer review reports and author responses from that submission.
Round 1
Reviewer 1 Report
It was a pleasure to review the manuscript entitled “Nicotinamide induced epigenetic rescue of T cell exhaustion” from Sara Alavi and colleagues.
In their manuscript, the authors aim to study an important topic: the metabolic prevention of exhaustion in T cells by modulating their metabolic fitness. Indeed, mitochondrial dysfunction in exhausted T cells has been reported by others, and enhancing mitochondrial fitness, for instance, using NAM, is a strategy followed to reinvigorate T cell functions (DOI: 10.1038/s41590-020-0793-3, https://doi.org/10.1016/j.celrep.2021.109516).
Using an in vitro model of T cell exhaustion, generated through repeated stimulation with CD3-CD28 beads, this study describes the effect of NAM supplementation or EZH2 inhibition in preventing T cell exhaustion.
The data described in this study show reduction in IR expression on the in vitro induced “exhausted T cells” in response to NAM, and a divergent effect of EZH2 inhibition.
While the topic is of high interest in the field, major weaknesses hamper conclusions made in this paper:
- The model used (repeated stimulation with anti-CD3/CD28 beads) is missing key information to demonstrate a T-cell exhaustion state (there is no mention on T-cell viability, proliferation capacity). Beside presence of IR, the levels of TOX and cytokines expression profiles are not consistent with an exhausted state.
- Cytokine profiles are not consistent with an exhaustion state, but with a chronic activation state as in Figure 2A, at day 9 after 4 rounds of CD3/CD28 stimulation, the percentage of IL2 positive cells detected by ICS is higher than control.
- Several key results are poorly reproducible as seen by levels of TOX that are increased consistently in Figure 3A at day 9 (after 3 CD3/CD28 stimulation) but then having different patterns at day 11 in Figure 5D. Overall lack of reproducibility and difficulty to cross-compare the same read-outs for key markers hamper the argument of the authors that this is indeed a controlled state of exhaustion.
- Overall there is no presence of statistics in the majority of the figures. The authors mentioned that the time course is depicted as a representative experiment, and in parallel that some of the data are not consistent among the different studies. We would suggest thus to present data as mean including SD, either of the different experiments or at least of technical replicates, in order to understand what is or not consistent and reproducible.
- Some of the main figures are of poor quality and very difficult to read and understand (see Figure 4D and 7B-D)
- Functional validation is missing in order to avoid overstatements
Introduction:
A better explanation of the role of EZH2 in T-cell exhaustion would be necessary.
Figure 1A
Model of repeated stimulation: repeated stimulation with CD3/CD28 beads has been used by others and was shown to induce an exhaustion-like state in T cells characterized by expression of IR and loss of cytokine secretion (DOI: 10.1007/978-1-0716-0171-6_6). Three to four rounds of stimulation are often used, as, after this, T cell can lose expression of CD28, then a subsequent stimulation with CD3-CD28 beads might induce T cell death. The current study provides no information regarding CD28 expression during the round of stimulation, cell number, and viability in control and treated cells. It is difficult to understand how NAM is affecting T cell beyond IR. Is it preventing T cell proliferation? Affecting T cell survival?
Figure 1:
If GSK343 inhibitor has no effect on inhibiting expression of IR, one suggestion would be to remove the data from the graphs and instead include error bars of the biological replicates and representation of the different experiments to see which data are the most reproducible. For example, it is stated that “ In one study there was a decrease in LAG3 in CD8+ T cells in the presence of NAM (Figure 1C)” → meaning that in 3 studies, there was no effect?
Figure 2: cytokine production should be downregulated upon T-cell exhaustion, while it seems that repeated stimulation with the CD3/CD28 beads enhances both IFNy and IL2. Why is that? Can the authors also evaluate extracellular medium to confirm reduction in cytokines production?
Figure 3: These figure results are unclear to understand as TOX and EZH2 are increasing, but NAM has an effect on the reduction of TOX but not on EZH2. The data can be described but functional validation would be required to avoid overstatement.
Figure 4: The conclusions of the graphs are not clear again as we are missing cell numbers and viability information.
We would expect the naïve compartment to be more represented at baseline.
TSNE is really of low quality, and again not easy to understand. It seems that there is a reduction on CD39+ in NAM treated. However, the overall number of cells look different. It would be good to have a proportion of IR in different conditions as well.
Figure 5: Here viability should be again demonstrated as such a high number of repeated stimulation might end up in inducing T cell death. No error bars on the graphs, no statistics again.
Cytokine profile is again very variable. Is it variability due to the different donors or a batch effect of the experiments?
TOX should be a marker of T cell exhaustion; why only 5-10% positive cells → are they truly exhausted? Just activated? Senescent?
Figure 6:
Although impact of NAM on ROS would be expected, data presented in this figure are again not reproducible, thus not sufficient to conclude the effect of NAM on ROS. Mitochondrial DNA measurement and confocal imaging should also be performed to confirm these findings.
Global conclusion and remark:
Investigation of the effect of NAM on T cell exhaustion is of high interest in the field, and using an in vitro model of exhausted T cells is a good strategy to dissect the mechanism behind it. While recent publication are validating the conclusions made in this paper, in that NAM can affect T cell state and enhance anti-tumor T cell fitness, the data presented in the above manuscript are mainly descriptive and missing statistical power and functional validation. Representation of some of the figures is of low quality.
Thus, I don’t believe that the paper should be accepted in this format. Data presented here are not sufficient to state that NAM impacts T cell exhaustion through epigenetic rescue.
Reviewer 2 Report
Alavi S. et al show that NAM supplementation during the induction of T cell exhaustion prevents the upregulation of inhibitory receptors and the suppression of cytokine secretion. Although there are some interesting observations, some statements are exaggerated and have to be revisited. For example, the title describes that NAM prevents T cell exhaustion through an epigenetic mechanism. However, the data provided in the report is insufficient for this statement as the direct link between NAM and EZH2 is poorly demonstrated.
Furthermore, Alavi S. et al describe two mechanisms (EZH2 and ROS) by which NAM might mediate the prevention of T cell exhaustion. Nonetheless, the evidence provided to sustain these two mechanisms is incomplete and the authors should consider to perform additional assays to strengthen these hypothesis:
- A key hypothesis stated by the authors is that NAM might induce EZH2 expression/activity and subsequently prevent the induction of TOX and other inhibitory receptors. However, the authors fail to show in their model that the modulation of EZH2 levels and activity (using GSK343) affects T cell exhaustion at all. Indeed, no significant differences are observed between control conditions and GSK343 conditions in terms of PD-1, TIM3, LAG3 and CD39 (Fig 1C) and TOX (Fig 3B). Moreover, cytokine levels are, if anything, increased by the loss of EZH2 (Fig. 2B). This suggests that perhaps the model proposed is not ideal to examine the link between NAM, EZH2 and exhaustion. The authors should prove the direct link between NAM and EZH2 and the possible implications in T cell exhaustion. For example, performing an experiment using activated T cells and induce its exhaustion using EZH2 inhibitors. The authors could show that NAM supplementation prevents EZH2-induced T cell exhaustion.
- The second mechanism supported by the authors is that NAM reduces ROS levels. However, the data provided is inconclusive. Indeed, NAM supplementation seems to increase ROS levels when compared to control and only rescues GSK-induced ROS increase in 1 of 2 experiments. Similarly, the results on MDR/MG ratios seem not sufficiently significant. More assays analysing the mitochondrial profile of these T cells is required to sustain this hypothesis.
Specific points:
- In Fig. 6B, the figure legend says NMN instead of NAM?
- Line 238-240 – “LAG expression in both CD4+ and CD8+ T cells and PD1 in CD4+ T cells were reduced suggesting that NAMs effects on these IRs involved EZH2 stimulation”. Overstatement. Indeed, pooled data (Fig. 1C) does not show this.
- Line 288 – “NAM did increase H3K27me3…”. Fig. 3 does not show this.
Reviewer 3 Report
This is an interesting study. Authors present convincing data about role of NAM in exhaustion. Authors must address following points.
- Line 247 please correct the first sentence.
- Line 263 should be Balkhi et al.,
- Line 278. Authors should clarify whether GSK treatment inhibited H3K27me3 levels. That is not clear from the description even though figure 3A- H3K27me3 panel shows that. Please mark these panels as A, B, C… for clarity. One of the legends in Figure 3A is shown in reverse order, correct that. In the figure 3A, there is no affect of GSK on Ezh2 protein as expected (less than 2-fold). I am not sure why authors state there is effect on the Ezh2 protein.
- One of the important points that authors should address whether there is an expansion of Tregs in their repeat cultures in absence and presence of all treatments. The experiment will rule out any Treg expansion due to IL2 recovery.
- Authors could further explore if NAM mediated effect depend on Ezh2 by knocking down Ezh2 in T cells.